🔓 | **Open Peer Review** | Antimicrobial Chemotherapy | Research Article

# Antimicrobial susceptibility and treatment of clinical cases of yersiniosis in Aotearoa | New Zealand

Lucia Rivas,[1] Jenny Szeto,[2] Juliet Elvy,[2,3] Kristin Dyet,[2] Jackie Wright,[1] Ernest Williams,[4] Brent Gilpin[1]

**ABSTRACT** Yersiniosis is the second most notified bacterial disease in Aotearoa | New Zealand (NZ). National clinical treatment guidelines for yersiniosis are available, but there is a lack of supporting antimicrobial susceptibility data for clinical isolates of *Yersinia* spp. and prescribing data for yersiniosis cases. Data were collected through interviews with 148 consenting individuals notified with yersiniosis. Forty-three percent ($n = 63$) of cases indicated antimicrobial use while symptomatic with yersiniosis, including two cases with dual diagnosis (Campylobacteriosis). Children under the age of 5 were predominantly notified with *Yersinia enterocolitica* (YE) biotype (BT) 2/3 (87%; 27/31) and reported significantly ($P = 0.008$) less antimicrobial use compared to adults (aged 20–70+ years). Antimicrobial use was significantly ($P = 0.006$) higher for cases reporting pre-existing gastrointestinal condition(s) and with YE BT 1A (26%; 12/47) compared to YE BT 2/3 (8%; 7/86). Eighty-six percent (44/51) of cases that indicated antimicrobial use identified the commencement date. Of these, 9% (4/44), 77% (34/44), and 14% (6/44) received an antimicrobial(s) as either empirical, directed, or empirical and directed treatment, respectively. Trimethoprim-sulfamethoxazole (49% [25/51]), ciprofloxacin (16% [8/51]), and doxycycline (10% [5/51]) were the most prescribed antimicrobials. Phenotypic antimicrobial susceptibility testing demonstrated clinical *Yersinia* spp. were susceptible to all antimicrobials commonly prescribed for the treatment of yersiniosis. Whole-genome sequence (WGS) analysis showed very few (1–3) antimicrobial resistance (AMR) genes within the *Yersinia* genomes. The results support the current antimicrobial prescribing recommendation for the treatment of yersiniosis in NZ and the utility of WGS to assess for AMR profiles in *Yersinia* spp.

**IMPORTANCE** This study demonstrates that currently almost half of yersiniosis cases interviewed receive antimicrobial treatment, with most prescriptions as directed therapy after a diagnosis has been made. This study also confirms the appropriateness of current treatment guidelines for the management of yersiniosis in NZ, and that most primary care practitioners in the study areas follow these recommendations. Phenotypic testing was well correlated with the genomic assessments of antimicrobial resistance (AMR). Both types of analysis indicated a low level of AMR for *Yersinia* spp. in New Zealand (NZ) compared to data from overseas studies. However, ongoing surveillance given the burden of yersiniosis and high rates of treatment in NZ is paramount to ensuring timely detection of emerging multi-drug resistance and to help devise evidence-informed interventions.

**KEYWORDS** *Yersinia*, antibiotic resistance, yersiniosis, gastrointestinal infection

Yersiniosis is a gastrointestinal infection caused by the bacteria *Yersinia enterocolitica* (YE) and less frequently, *Yersinia pseudotuberculosis* (YP). Symptoms in humans include diarrhea, vomiting, and abdominal pain. This infection can also give rise to

**Peer Reviewer** Luís Augusto Nero, University of Vicosa, Vicosa, Brazil

Address correspondence to Brent Gilpin, Brent.Gilpin@esr.cri.nz.

The authors declare no conflict of interest.

See the funding table on p. 13.

uncommon clinical manifestations and sequelae, and sepsis may occur in immunocompromised individuals (1).

Yersiniosis is a notifiable disease in Aotearoa | New Zealand (NZ) and is the second most frequently notified bacterial disease. Since 2015, the rates of yersiniosis in NZ have significantly increased with a peak of 1,410 cases in 2021 (27.5 cases per 100,000 population), stabilizing in 2022 at 1,294 cases (25.3 cases per 100,000 population) (2–4). The current rate of yersiniosis observed in NZ is high compared to other industrialized countries that ranging from 0.2 to 1.9 cases per 100,000 (2, 4–7). The predominance of YE biotype (BT) 4 is reported in other countries where surveillance and typing are available (5, 8). In NZ, a shift from the predominance of YE BT 4 to YE BT 2/3 among clinical cases has been observed since 2012; however, the reasons why this change occurred remain unclear (9). Also, YE BT 1A, a BT that is considered non-pathogenic due to the lack of classical YE virulence determinants is not notifiable in many countries, yet is observed as the second most common YE BT in notified cases in NZ (10, 11). A NZ case-control study on yersiniosis reported raw pork as a significant risk factor for exposure and infection to YE BT 2/3, but not YE BT 1A, indicating the etiology of different YE biotypes in NZ is complex (9).

In NZ, antimicrobial prescribing guidance is available to primary care practitioners for the management of yersiniosis (12). These guidelines recommend antimicrobials only for cases with severe symptoms or who are immunocompromised, with doxycycline the first-choice antimicrobial, followed by trimethoprim-sulfamethoxazole or ciprofloxacin. However, it is unclear how frequently antimicrobials are prescribed for the treatment of yersiniosis in NZ, and if so, which antimicrobials are used or their relative effectiveness.

In NZ, yersiniosis cases are recorded within a nationally coordinated surveillance system (EpiSurv) and clinical isolates are required to be referred from diagnostic laboratories to the Institute of Environmental Science and Research (ESR) Enteric Reference Laboratory for epidemiological typing (1). ESR undertakes the surveillance of notifiable diseases in NZ, including yersiniosis on behalf of the Ministry of Health. However, phenotypic antimicrobial susceptibility testing (AST) is not routinely performed by NZ laboratories, and the rates of antimicrobial resistance (AMR) for NZ and international yersiniosis isolates are lacking (4, 13). Increasing the use of whole-genome sequence (WGS) analysis for routine surveillance of pathogens presents an opportunity to provide additional information about the presence of AMR genes or the "resistome," which may have an application to empiric treatment recommendations (14). However, the correlation between phenotypic susceptibility data and the resistome requires further evaluation.

The aim of the current study was to determine the antimicrobial susceptibility of *Yersinia* spp. obtained from yersiniosis cases identified through a previous case-control study conducted in 2021–2022 (9); to compare phenotypic and genomically inferred AMR results; and to evaluate antimicrobial use among yersiniosis cases, including initial symptoms and a pre-existing gastrointestinal condition(s) reported by the case.

## MATERIALS AND METHODS

### Case recruitment and questionnaire

Confirmed cases of yersiniosis were defined as a clinically compatible illness accompanied by laboratory definitive evidence of either (a) isolation of YE or YP from blood or feces, or (b) detection of *Yersinia* spp., nucleic acid from feces (1). Diagnostic primary laboratories report testing results through a direct laboratory notification system, and confirmed cases are notified to EpiSurv.

A case-control study of yersiniosis cases reported from the Canterbury and Wellington regions was performed during 2021–2022 (9). All yersiniosis cases were initially eligible for inclusion in the study, regardless of age, gender, and ethnicity. The routine protocol for public health investigation is that a parent or guardian is interviewed on behalf of a notified case that is under 16 years of age. The parent or caregiver must reside in the

same household as the case. For the case-control study, cases aged 7–15 years old were provided the opportunity to participate in the interview or they could assent for their parent or caregiver to be interviewed on their behalf.

A case was excluded from the study if, during the interview, it was determined that the case was either asymptomatic, was not able to communicate (personally or through a proxy) in English, was overseas within 10 days prior to illness onset, or was resident in an institution or equivalent (i.e., did not have access to a personal telephone) at the date of notification.

Following completion of the interview, all eligible cases (or their assented proxy reporters) were provided information regarding the study (via email/postal mail and telephone) and were subsequently contacted to seek consent to use their anonymized responses to the questionnaire for the purpose of the study. Cases were requested to participate in a follow-up questionnaire as a part of the case-control study, which aimed to collect the following information:

1. Initial symptoms experienced when becoming unwell with yersiniosis include diarrhea (three or more loose bowel motions in a 24-h period), vomiting, or the feeling of vomiting, abdominal pain, or stomach cramps or pain.
2. Any pre-existing gastrointestinal condition(s) and whether the above symptoms were being experienced prior to becoming ill with yersiniosis.
3. Hospitalization and admittance to hospital due to yersiniosis.
4. Use of an antimicrobial prescribed for yersiniosis, including date commenced and which antimicrobial. Of those cases that were able to identify a commencement date, a comparison was undertaken using the case notification date on EpiSurv. Those cases with commencement dates before the notification date were presumed to be treated empirically for gastrointestinal symptoms; and those who commenced after were presumed to be directed therapy for yersiniosis.

Information collected for this study was directly obtained from the case interviews only, and access to medical records to verify the information provided, including clinical or antimicrobial dispensing, was not possible. Consultation with the prescribing primary care practitioners was also not possible to elucidate the reasons for the prescription for an alternative antimicrobial for a case, especially as pre-existing gastrointestinal condition(s) were also reported among the cases.

Each case provided verbal consent to the public health interviewer for the use of anonymized responses for the purpose of the current study. All cases were interviewed by the same interviewer. Ethics approval for the study was obtained from the Southern Health and Disability Ethics Committee (HDEC; reference 21/STH/84).

The Pearson's chi-squared test using R software was used to assess statistical significance within selected data sets (15).

## Bacterial isolates and genomic assessment

A total of 109 *Yersinia* spp. clinical isolates obtained from notified yersiniosis cases that participated in the case-control study (9) were selected for phenotypic AST and WGS (Table S1). For WGS, all isolates were grown on Tryptone Soya Agar (Fort Richard) at 28°C for 24 ± 2 h. This was then used to inoculate 1 mL TE buffer to a density of 2–4 McFarland Standard, which was then heat inactivated at 65°C for 1 h. These suspensions were stored at 4°C until genomic DNA extraction using the Chemagic 360 extraction platform (PerkinElmer, Waltham, MA, USA). DNA quality and concentration were performed using PicoGreen (Quant-iT; Thermo Fisher Scientific). Sequencing libraries containing 1 ng of DNA were prepared using Nextera XT chemistry (Illumina, San Diego, CA, USA) for 150 bp pair-end sequencing on an Illumina NextSeq sequencer, according to the manufacturer's recommendations (Illumina).

Raw sequence reads for all isolates analyzed in this study are available on the National Centre for Biotechnology Information (NCBI) short read archive (SRA) with BioProject number PRJNA 1142067.

Initial sequence quality and species identification using the Illumina data were determined using the Nullarbor version 2 pipeline (16). The multi-locus sequence type (ST) was inferred using WGS data, applying the McNally scheme for YE and the Achtman scheme for YP (17, 18). Isolates to be included in the current study were selected to cover a range of different *Yersinia* species and YE BTs, as well as seven-gene multi-locus STs as previously observed (9). This included 79 isolates that corresponded to notified yersiniosis cases that consented to the follow-up interview as outlined above (17, 18). AMR genes were inferred using the WGS *abritAMR* pipeline with AMRFinderPlus (19, 20).

## AST

Isolates were cultured on Tryptic Soy Agar prior to AST, which was performed using the microdilution susceptibility method with Sensititre NARMS Gram Negative plates (CMV3AGNF, ThermoFisher Scientific; Waltham, MA, USA) as per the Clinical and Laboratory Standards Institute (CLSI) guidelines (21) and the manufacturer's instructions. This method provides a gold-level standard for phenotypic susceptibility testing of bacterial isolates (22–24). The standardized Sensititre plate contained 14 antimicrobials and was selected for use as it contained the following antimicrobials relevant for human clinical use in NZ: cefoxitin, azithromycin, tetracycline, ceftriaxone, amoxicillin-clavulanic acid (2:1 ratio), ciprofloxacin, gentamicin, trimethoprim-sulfamethoxazole and ampicillin. In addition, meropenem (ThermoFisher) and quinupristin-dalfopristin (Bio-Rad; Hercules, CA, USA) were tested using the disk diffusion method and as per guidelines of the European Committee on Antimicrobial Susceptibility Testing (EUCAST). Quality control strains, *Escherichia coli* ATCC 25922, *E. coli* ATCC 35218, *Enterococcus faecalis* ATCC 29212, and *Staphylococcus aureus* ATCC 29213, were included in every batch of Sensititre plates, including off-plate dilution when necessary. Results were only reported if the control strains were within range, as specified by EUCAST or CLSI.

The concentration range, interpretive standards, and breakpoints used are summarized in Table S2.

## RESULTS

A total of 148 eligible and consenting cases were interviewed in this study (Table 1). The median number of days between case notification and follow-up interviews with the questions specific to the current study was 7 days. The proportion of cases interviewed based on YE BT was consistent with the overall notification numbers observed for the case-control study and nationally (9). Of the cases that were interviewed in the current study, 58% (86/148) of cases were notified with YE BT 2/3, followed by 32% (47/148) with YE BT 1A. Cases notified with YE BT4 and YP represented 3–4% of the cases interviewed (four and six cases, respectively). The number of children under the age of 5 and notified with YE BT 2/3 was higher (87%; 27/31) than those notified with YE BT 1A (6%; 2/31). In contrast, 57% (49/86) of interviewed cases aged 20–70+ years were notified with YE 2/3, while 47% (40/86) were identified with YE BT 1A.

Forty-three percent (63/148) of interviewed yersiniosis cases indicated antimicrobial use while symptomatic with yersiniosis (Table 1). Five additional interviewed cases had indicated antimicrobial use for another health condition and not for yersiniosis, and three cases were prescribed an antimicrobial but chose not to take it. These cases were recorded as not taking an antimicrobial for the treatment of yersiniosis and not included within the values presented in Table 1. Seven cases were reported with dual diagnoses (including five with campylobacteriosis; two of whom indicated antimicrobial use and included in Table 2). One case each was also diagnosed with *Aeromonas* and Shiga toxin-producing *E. coli*, but neither of these cases indicated antimicrobial use.

The number of cases that indicated antimicrobial use for yersiniosis was significantly ($P = 0.008$) different across age groups, with children under 5 years of age reported to

**TABLE 1** Overview of age and gender of yersiniosis cases interviewed and antimicrobial use for the treatment of yersiniosis

| Age group (years) | *Yersinia* species and biotype (BT)(%)[a] | | | | | Total cases in age group (%)[b] | No. of cases that indicated antimicrobial use in age group (%)[c] |
|---|---|---|---|---|---|---|---|
| | YE BT 1A | YE BT 2/3 | YE BT4 | YP | *Yersinia* other | | |
| <1–4 | 2 | 27 (31%) | | 1 | 1 | 31 (21%) | 6 (19%) |
| 5–9 | 2 | 3 | | 2 | | 7 (5%) | 1 |
| 10–14 | 1 | 3 | | 1 | | 5 (3%) | 2 |
| 15–19 | 2 | 4 | 1 | 2 | | 9 (6%) | 4 |
| 20–29 | 7 (15%) | 14 (16%) | | 1 | | 22 (15%) | 6 (27%) |
| 30–39 | 6 (13%) | 8 (9%) | 1 | | 2 | 17 (11%) | 10 (59%) |
| 40–49 | 6 (13%) | 8 (9%) | | | | 14 (9%) | 6 (42%) |
| 50–59 | 8 (17%) | 5 (6%) | 2 | | 1 | 16 (11%) | 10 (63%) |
| 60–69 | 10 (21%) | 6 (7%) | | | | 16 (11%) | 11 (69%) |
| 70+ | 3 | 8 (9%) | | | | 11 (7%) | 7 (70%) |
| **Gender** | | | | | | **Total cases in gender group (%)[b]** | **No. of cases that indicated antimicrobial use in gender group (%)[d]** |
| Female | 27 (57%) | 51 (59%) | 0 | 3 | 3 | 84 (57%) | 40 (48%) |
| Male | 20 (43%) | 35 (41%) | 4 | 4 | 1 | 64 (43%) | 23 (36%) |
| Total (%)[b] | 47 (32%) | 86 (58%) | 4 | 7 (5%) | 4 | 148 | 63 (43%) |
| No. of cases that indicated antimicrobial use within *Yersinia* species and BT group (%)[a] | 26 (55%) | 32 (37%) | 1 | 2 | 2 | | |

[a]*Yersinia enterocolitica* (YE) and biotype (BT). *Yersinia pseudotuberculosis* (YP). Other *Yersinia* includes case samples that were screen positive for YE at the primary diagnostic laboratory, but BT was not confirmed. Percentage calculated based on total number of cases within a category (for five or more cases only).
[b]Percentage calculated for each age group, gender and antimicrobial use (with five or more cases only) based on total number of cases ($n = 148$).
[c]Percentage calculated for each age group (with five or more cases only) based on total number of cases in the age group.
[d]Percentage calculated for each gender group.

have significantly ($P = 0.003$) less antimicrobial use than cases within the age groups of 20–70+ years (Table 1). There was no statistical difference ($P = 0.15$) in antimicrobial use between female and male cases (48%; 40/84 and 36%; 23/64) (Table 1). Antimicrobial use was significantly ($P = 0.006$) higher for cases with YE BT 1A (26%; 12/47) compared to YE BT 2/3 (8%; 7/86). The number of cases that self-reported a pre-existing gastrointestinal condition(s) was significantly ($P = 0.004$) higher for cases confirmed with YE BT 1A (49%; 23/47) than cases with YE BT 2/3 (24%; 21/86). However, the number of cases of YE BT 1A that indicated antimicrobial use did not differ significantly ($P = 0.8$) between those with or without pre-existing condition(s) (12/47; 26% and 9/47; 19%, respectively).

Information on the selected initial symptoms experienced by the interviewed yersiniosis cases is outlined in Table 2. Diarrhea, abdominal pain, and severe stomach pain (not as a pre-existing symptom) were more frequently reported (ranging from 51% to 75%) among cases compared to vomiting (20%; $P < 0.001$). Eighty-three percent (123/148) reported experiencing more than one symptom (data not shown). Excluding those with existing diarrhea, vomiting, or abdominal pain, there was no statistical ($P \geq 0.09$) difference in symptoms reported by cases notified with YE 2/3 or YE BT 1A. Cases that provided information on the number of days they experienced each symptom reported symptom duration of between 1 and 60 days. Some cases reported prolonged and intermittent symptoms. Antimicrobial use was reported in 34–44% of cases for each of the symptom categories (not as pre-existing symptoms).

Thirty-seven out of 148 interviewed cases (25%) reported attending the hospital emergency department while they were ill, of which 25 (17% of all cases) were subsequently hospitalized for yersiniosis. Of the 25 hospitalized cases, 13 (52%) indicated a pre-existing gastrointestinal condition(s). Twenty of the hospitalized case isolates (80%) were confirmed as YE BT 2/3, while one was YE BT 4, three were YE BT 1A (one had a dual diagnosis with campylobacteriosis), and one case was YP. Severe abdominal

**TABLE 2** Overview of selected initial symptoms and antimicrobial use reported for interviewed yersiniosis cases

| Symptom | *Yersinia* type[a] | Number of cases that responded (%)[b] | | | | Total | Day range of symptoms[c] |
|---|---|---|---|---|---|---|---|
| | | Yes | Exist | Exist but worse | No | | |
| Diarrhoea | YE BT 1A | 21 (46%) | 6 (13%) | 11 (24%) | 8 (17%) | 46 | 1 day to >60 days |
| | YE BT 2/3 | 68 (79%) | | 1 | 17 (20%) | 86 | |
| | YE BT 4 | 4 (100%) | | | | 4 | |
| | YP | 5 | | | 2 | 7 | |
| | *Yersinia* other | 1 | | | 3 | 4 | |
| | Total no. of cases (%) | 99 (67%) | 6 (4%) | 12 (8%) | 30 (20%) | 147 | |
| | No. of cases that indicated antimicrobial use (%) | 34 (34%) | 5 (83%) | 8 (67%) | 15 (50%) | 62 | |
| Vomiting | YE BT 1A | 14 (29%) | 1 | | 32 (68%) | 47 | 1–4 days |
| | YE BT 2/3 | 15 (17%) | 1 | | 70 (80%) | 86 | |
| | YE BT 4 | 1 | | | 3 | 4 | |
| | YP | | | | 7 (100%) | 7 | |
| | *Yersinia* other | 1 | | | 3 | 4 | |
| | Total no. of cases (%) | 31 (20%) | 2 | | 115 (78%) | 148 | |
| | No. of cases that indicated antimicrobial use (%) | 13 (42%) | 1 | | 49 (42%) | 63 | |
| Abdominal pain | YE BT 1A | 27 (57%) | 4 | 10 (21%) | 6 (13%) | 47 | 1–32 days |
| | YE BT 2/3 | 69 (81%) | | 1 | 15 (18%) | 85 | |
| | YE BT 4 | 3 | | 1 | | 4 | |
| | YP | 7 (100%) | | | | 7 | |
| | *Yersinia* other | 4 | | | | 4 | |
| | Total no. of cases (%) | 110 (75%) | 4 | 12 (8%) | 21 (14%) | 147 | |
| | No. of cases that indicated antimicrobial use (%) | 43 (39%) | 3 | 8 (76%) | 9 (42%) | 63 | |
| Severe abdominal pain | YE BT 1A | 20 (67%) | | 4 | 23 (49%) | 47 | 1–60d |
| | YE BT 2/3 | 49 (58%) | | 1 | 34 (40%) | 84 | |
| | YE BT 4 | 2 | | | 2 | 4 | |
| | YP | 3 | | | 4 | 7 | |
| | *Yersinia* other | 1 | | | 3 | 4 | |
| | Total no. of cases (%) | 75 (51%) | | 5 (3%) | 66 (45%) | 146 | |
| | No. of cases that indicated antimicrobial use (%) | 33 (44%) | | 5 (100%) | 33 (50%) | 63 | |

[a]Yersinia enterocolitica (YE) and biotype (BT). *Yersinia pseudotuberculosis* (YP). *Yersinia* other includes case samples that were screen positive for YE at the primary diagnostic laboratory, but BT was not confirmed. Percentages calculated based on total number of cases (*n* = 148). Calculated for case numbers >5 only.
[b]Cases were asked for each symptom whether a) "Yes—they experienced as a new symptom," assigned as "Yes," "Yes, but had the symptom or medical condition before and the symptom stay the same after yersiniosis," assigned as "Exist," "Yes, but had the symptom or medical condition before and the symptom got worse after yersiniosis," assigned as "Exist but worse," or "No, did not experience the symptom," assigned as "No." Percentage calculated based on the total number of cases for each *Yersinia* type that provided an answer for each symptom. Calculated for case numbers >5 only.
[c]The number range of days self-reported by cases when provided.
[d]Percentage calculated based on the total number of cases interviewed who answered each question (*n* = 146–148). Calculated for case numbers >5 only.

pain and suspected appendicitis were commonly noted among reported reasons for hospitalization.

Of the 63 interviewed cases that indicated antimicrobial use, 44 cases (70%) were able to identify the start date (Table 3). Of these cases, 77% (34/44) used a single antimicrobial for the directed treatment of yersiniosis, compared to 10/44 (23%) treated empirically. Antimicrobial use among directed therapy cases started on average, 3 days (ranging 0–17 days) following diagnosis. This included nine caseshospitalized due to yersiniosis. The most common single antimicrobial reported was trimethoprim-sulfamethoxazole (49% [25/51]), while ciprofloxacin and doxycycline were the second and third most common (16% [8/51] and 10% [5/51]), respectively (Table 3). Trimethoprim, azithromycin, norfloxacin, or amoxicillin were also reportedly used among five cases.

Of the 10 cases treated empirically, 4 were unable to name the antimicrobial, while another 4 cases (all YE BT 2/3) were hospitalized and given an intravenous antimicrobial

**TABLE 3** Overview of antimicrobials reported to be taken by 63 interviewed yersinosis cases

| Category | Number of cases based on *Yersinia* species,[a] biotype (BT)[b] | | | | | | Number of cases that indicated antimicrobial use[c] | | | | Comments |
|---|---|---|---|---|---|---|---|---|---|---|---|
| | YE BT 1A | YE BT 2/3 | YE BT 4 | YP | Yersinia other | Total (%)[b] | Able to identify start date | Empirical treatment only | Direct treatment only | Empirical and direct treatment | |
| Number of cases interviewed that identified the antimicrobial(s) used | 20 | 27 | 1 | 1 | 3 | 51 | 44 | 4 | 34 | 6 | |
| Trimethoprim-sulfamethoxazole | 9 | 12 | 1 | 1 | 3 | 25 (49%) | 22 | 1 | 21 | 0 | Four cases hospitalised and had direct treatment |
| Ciprofloxacin | 4 | 4 | | | | 8 (16%) | 8 | 0 | 8 | 0 | Four cases hospitalised and had direct treatment |
| Doxycycline | 4 | 1 | | | | 5 (10%) | 3 | 1 | 1 | 0 | One case hospitalised and had direct treatment |
| Intravenous (unknown type) (2) ciprofloxacin | | 2 | | | | 3 (6%) | 2 | 0 | 0 | 2 | All cases hospitalised. Empirical use (1) and then direct treatment use (2) |
| Azithromycin | | 3 | | | | 2 (4%) | 3 | 0 | 3 | 0 | One case allergic to sulphur antimicrobials |
| Norfloxacin | | 1 | | | | 1 (2%) | 1 | 0 | 1 | 0 | |
| Intravenous (amoxicillin-clavulanic acid) (2) ciprofloxacin | | 1 | | | | 1 (2%) | 1 | 0 | 0 | 1 | Case hospitalised. Empirical use of (1) and then direct treatment use of (2) |
| Amoxicillin | | 1 | | | | 1 (2%) | 1 | 1 | 0 | 0 | Case hospitalised |
| Trimethoprim | | 1 | | | | 1 (2%) | 0 | 0 | 0 | 0 | |
| Metronidazole (2) azithromycin | 1 | | | | | 1 (2%) | 1 | 0 | 0 | 1 | Case hospitalised, empirical use of (1). Dual diagnosis Campylobacteriosis and then direct treatment use of (2) |
| Metronidazole (2) trimethoprim-sulfamethoxazole | 1 | | | | | 1 (2%) | 1 | 1 | 0 | 0 | Both antimicrobials are empirical |
| Amoxicillin-clavulanic acid (2) trimethoprim-sulfamethoxazole | 1 | | | | | 1 (2%) | 1 | 0 | 0 | 1 | Case hospitalised, empirical (1). Symptoms did not resolve, and then direct treatment use (2) |
| Trimethoprim-sulfamethoxazole (2) unknown antimicrobial | 1 | | | | | 1 (2%) | 1 | 0 | 1 | 0 | Dual diagnosis: Campylobacteriosis. Direct treatment (1). Symptoms did not resolve, so then used (2) |

**TABLE 3** Overview of antimicrobials reported to be taken by 63 interviewed yersinosis cases (*Continued*)

| Category | Number of cases based on *Yersinia* species,[a] biotype (BT) | | | | | Number of cases that indicated antimicrobial use[c] | | | | Comments |
|---|---|---|---|---|---|---|---|---|---|---|
| | YE BT 1A | YE BT 2/3 | YE BT 4 | YP | *Yersinia* other | Total (%)[b] | Able to identify start date | Empirical treatment only | Direct treatment only | Empirical and direct treatment |
| Unable to identify the antimicrobial taken | 6 | 5 | | 1 | | 12 | 8 | 4 | 4 | 0 |

[a]*Yersinia enterocolitica* (YE) and biotype (BT). *Yersinia pseudotuberculosis* (YP). *Yersinia* other includes case samples that were screen positive for YE at the primary diagnostic laboratory, but BT was not confirmed.
[b]Percentage calculated based on total number of cases that were able to identify the antimicrobial(s) used (*n* = 51).
[c]If the case had an EpiSurv notification date for yersiniosis before or after the date of starting an antimicrobial then they were recorded as "empirical treatment" or "direct treatment," respectively.

(1 case identified amoxicillin-clavulanic acid, and the remaining 3 cases could not identify the antimicrobial provided). All four of these cases reported subsequent use of ciprofloxacin for directed treatment of yersiniosis.

Three cases were prescribed an empirical antimicrobial ineffective for yersiniosis, with two cases receiving metronidazole and another receiving amoxicillin. Two of these cases were switched to an appropriate directed therapy once the result was available (trimethoprim-sulfamethoxazole and azithromycin). One case was empirically treated with amoxicillin-clavulanic acid but was subsequently prescribed trimethoprim-sulfame-thoxazole as directed therapy due to ongoing symptoms. The AST results showed *Yersinia* spp. isolates were highly susceptible to the antimicrobials recommended for the treatment of yersiniosis and other clinically relevant antimicrobials commonly used in NZ (Table 4; Table S1).

WGS analysis resulted in very few (1–3) AMR genes detected among the *Yersinia* genomes (Table 4). All YE isolates were resistant to ampicillin, which correlated with the full or partial detection of the *blaA* gene observed in the genomes. All YE BT 1A and YE BT 2/3 ST14 isolates were phenotypically resistant to amoxicillin-clavulanic acid and cefoxitin, but variability in resistance to cefoxitin was observed for six YE BT 2/3 ST12 isolates. This included an isolate of YE BT 2/3 ST 12 that had $bla_{TEM-1}$, as well as a partial detection of *blaA*, and was phenotypically sensitive to cefoxitin but resistant to amoxicillin-clavulanic acid. All YE BT 4 isolates ($n = 6$) were phenotypically susceptible to amoxicillin-clavulanic acid and cefoxitin, despite the detection of the entire *blaA* gene. A single isolate in the data set (YE BT 2/3, ST12) had *tetA* inferred, which correlated to a single detection of phenotypic resistance to tetracycline observed for this isolate. No AST phenotype (quinupristin/dalfopristin resistance) was observed to correlate with the presence of the *vatF* gene, which was observed either as complete or partially in all YE isolates tested. All isolates representing other *Yersinia* spp., including YP, had no AMR genes detected and were susceptible to all antimicrobials tested.

## DISCUSSION

In NZ, yersiniosis is considered to be self-limiting, and antimicrobial prescribing guidelines advise that antimicrobials are only prescribed to children and adults with severe symptoms or who are immunosuppressed (12). Although children under five years of age represented the age group with the highest number of yersiniosis cases, particularly for YE BT 2/3, this age group had a low level of antimicrobial use compared to other age groups. In contrast, antimicrobial use was significantly higher among cases notified with YE BT 1A, which were predominantly adults aged 20–70+ years old.

Overall, diarrhea and abdominal pain, including severe abdominal pain, were commonly reported symptoms among the interviewed cases that indicated no pre-existing gastrointestinal condition(s), but unlike the Finnish study (25), no significant differences were observed between reported symptoms and YE biotypes. However, it is of interest that 80% of hospitalized cases in the current study were reported as YE BT 2/3, with severe abdominal pain commonly noted among the self-reporting comments. This type of abdominal pain (particularly of the right lower quadrant of the abdomen) is similar to the pain experienced with appendicitis but is instead associated with mesenteric adenitis, which can be caused by YE (26). A German study reported that diarrhea, abdominal pain, and fever were more frequent in yersiniosis patients positive for serotype O:3 (which correlates with BT 4) compared to other non-O:3 (including O:9, O:5, 27 [which both correlate to BT 2/3], or non-specified serotypes). The high rate of hospitalization (27%; 153/571) was also significantly associated with pain in the lower right abdomen (27). In contrast, a Swiss study reported no significant difference in symptoms or hospitalization between patients with YE BT 1A ($n = 23$) or pathogenic YE (BTs 2, 3, and 4; $n = 43$) (28). The smaller number of case participants in the current study ($n = 148$) and the Swiss study ([28]; $n = 66$) may explain why a statistical significance between case symptoms and YE BT was not observed in contrast to other studies (Finnish study; $n = 295$ [25] and German study; $n = 571$ [27]).

**TABLE 4** Antimicrobial susceptibility testing and antimicrobial resistance gene results for *Yersinia* spp. isolated from clinical cases

| *Yersinia* species, biotype, ST[a] (number of isolates) | Antimicrobial tested and number of isolates[b] | | | | | | | | | | WGS AMR genes[c] | | |
|---|---|---|---|---|---|---|---|---|---|---|---|---|---|
| | FOX | AZI | TET | AXO | AUG2 | CIP | GEN | SXT | AMP | MER | Streptogramin | NSBL | Tetracycline |
| | S, I, R | S, R | S, R | S, R | S, R | S, R | S, R | S, R | S, R | S, R | *vatF*, *vatF* partial | *blaA*, *blaA* partial | *tetA* |
| YE BT 1A (*n* = 52) | 0, 0, **52** | 52, 0 | 52, 0 | 52, 0 | 0, **52** | 52, 0 | 52, 0 | 52, 0 | 0, **52** | 52, 0 | 0, 52 | 14, 38 | 0 |
| YE BT 2/3 (*n* = 42) | 2, 4, **36** | 42, 0 | 41, **1** | 42, 0 | 1, **41** | 42, 0 | 42, 0 | 42, 0 | 0, **42** | 42, 0 | 42, 0 | 0, 42** | 1 |
| ST12 (*n* = 37) | 2, 4, **31** | 37, 0 | 36, **1** | 37, 0 | 1, **36** | 37, 0 | 37, 0 | 37, 0 | 0, **37** | 37, 0 | 37, 0 | 0, 37** | 1 |
| ST14 (*n* = 5) | 0, 0, **5** | 5, 0 | 5, 0 | 5, 0 | 0, **5** | 5, 0 | 5, 0 | 5, 0 | 0, **5** | 5, 0 | 5, 0 | 0, 5 | 0 |
| YE BT 4 (*n* = 6) | 6, 0, 0 | 6, 0 | 6, 0 | 6, 0 | 6, 0 | 6, 0 | 6, 0 | 6, 0 | 0, **6** | 6, 0 | 6, 0 | 6, 0 | 0 |
| YP (*n* = 7) | 7, 0, 0 | 7, 0 | 7, 0 | 7, 0 | 7, 0 | 7, 0 | 7, 0 | 7, 0 | 7, 0 | 7, 0 | 0, 0 | 0, 0 | 0 |
| YF (*n* = 1) | 0, 0, **1** | 1, 0 | 1, 0 | 1, 0 | 0, **1** | 1, 0 | 1, 0 | 1, 0 | 0, 1 | 1, 0 | 0, 0 | 0, 0 | 0 |
| YH (*n* = 1) | 1, 0, 0 | 1, 0 | 1, 0 | 1, 0 | 1, 0 | 1, 0 | 1, 0 | 1, 0 | 1, 0 | 1, 0 | 0, 0 | 0, 0 | 0 |
| **Total no. (*n* = 109)** | 16, 4, 89 | 109, 0 | 108, **1** | 109, 0 | 15, **94** | 109, 0 | 109, 0 | 109, 0 | 8, **101** | 109, 0 | 48, 52 | 20, 80 | 1 |

[a]*Yersinia enterocolitica* (YE), biotype (BT) and multi-locus sequence type (ST) assigned for YE as per the scheme of McNally (12). *Y. pseudotuberculosis* (YP), *Y. frederiksenii* (YF), and *Y. hibernica* (YH).
[b]Antimicrobials tested within NARMS Gram-Negative Sensititre plate including cefoxitin (FOX), azithromycin (AZI), tetracycline (TET), ceftriaxone (AXO), amoxicillin–clav 2:1 ratio (AUG2), ciprofloxacin (CIP), gentamicin (GEN), nalidixic acid (NAL), ceftiofur (XNL), trimethoprim- sulfamethoxazole (SXT), ampicillin (AMP), and streptomycin (STR). Disk diffusion assay performed for meropenem (dMER) and ceftazidime (dCAZ). Interpretative standards and breakpoints assessed as per Table S2, and assigned as sensitive (S), intermediate (I) or resistant (R). Numbers in bold indicate where resistance was observed.
[c]Antimicrobial resistance (AMR) genes inferred from whole-genome sequencing data. Narrow-spectrum beta lactamase (NSBL). *Partial recovery of genes from contigs reported when there is between 50% and 90% of the gene present with >90% identity to a protein in the database. **1 YE ST12 had *bla*TEM-1 and partial recovery of *blaA*.

In addition, information collected for this study was directly obtained from the case interviews only, and access to medical records to verify the information provided was not possible. This collection of information relied on the case's memory recall and accuracy, which can be deemed as a limitation of the study, especially with the time delay between case notification and interviewing (medium seven days). Consultation with the prescribing practitioners was also not possible to elucidate the reasons for the prescription for a case (including pre-existing health condition(s)) and confirmation of the antimicrobial treatment prescribed. Insufficient data were collected to inform on how a case was prescribed an antimicrobial, but some case comments did indicate a follow-up by their primary care practitioner to inform them of their yersiniosis diagnosis, and additional appointment(s) to obtain the prescription, or no prescription was provided if case symptoms were improving.

Self-reporting of a pre-existing gastrointestinal condition(s) together with greater uncertainty or inability to identify a start date of when symptoms associated with yersiniosis began predominated among the YE BT 1A group compared to YE BT 2/3 (data not shown); an observation reported in a Finnish study as well (25). It may be hypothesized that the higher antimicrobial use observed among the YE BT 1A group could be due to cases experiencing prolonged (pre-existing) gastrointestinal condition(s), which prompted further investigatory testing and subsequent yersiniosis diagnosis and antimicrobial treatment. The role of YE BT 1A causing human disease remains controversial because BT lacks the classical YE virulence determinants (10, 11). Other studies have hypothesized whether at least a subset of YE BT 1A may be associated with gastrointestinal disease, especially in immunocompromised individuals, as well as alternative virulence mechanisms causing different symptom outcomes (11, 25, 29).

In the current study, trimethoprim-sulfamethoxazole, ciprofloxacin, and doxycycline were the most commonly reported antimicrobials used among the interviewed yersiniosis cases. The majority (77%) of cases that reported antimicrobial use (and provided a commencement date) received directed therapy for yersiniosis in response to the laboratory report. This study indicates that the overall NZ health guidelines are largely adhered to by prescribing practitioners. Only a minority of cases received empirical treatment for acute diarrhea of unknown pathology (12, 30). However, approximately 10% of treated cases were given an inappropriate antimicrobial for yersiniosis.

Our data confirmed that all clinical *Yersinia* spp. were susceptible to trimethoprim-sulfamethoxazole and ciprofloxacin, and all but one were susceptible to tetracycline. The 2024 EUCAST guidelines for clinical AST interpretation state that tetracycline can be used to predict doxycycline susceptibility for the treatment of YE infections (31). The results are consistent with expected AST phenotypes for YE as outlined by EUCAST and other international studies (32–39).

The detection of very few AMR genes within the *Yersinia* spp. genomes supported the AST findings in the current study. Resistance of YE to ampicillin observed in the current study has been frequently reported internationally and correlated to the detection of *blaA* beta-lactamases associated with the ability of YE to hydrolyze penicillin and cephalosporins (40–42). The current study observed differences in the susceptibility to amoxicillin-clavulanic acid and cefoxitin between YE BTs, which has also been previously reported in the literature (34, 42–44). Other studies have reported correlations between the presence of the *blaA* (entire or partial) and resistance to cefoxitin, amoxicillin-clavulanic acid, and ampicillin (45, 46), which was not observed in the current study. Genomic heterogeneity of *blaA* in YE BT1A has been previously reported (40). In addition, one YE BT 2/3, ST12 was observed to possess the $bla_{TEM-1}$ gene, which encodes the TEM β-lactamase enzyme. This gene has been previously observed at higher frequencies (40–85%) in YE from raw meat and dairy isolates from South Africa and Egypt, respectively, and is reported to also correlate to ampicillin resistance (47, 48). The findings support previous observations that the regulation of beta-lactamases in YE is complex and can

be expressed differently in different biotypes (49) and confirm the preferential use of alternative (i.e., non-beta lactam) agents.

Using WGS data, all YE genomes were found to possess the complete or partial *vatF* gene. This gene has been described within the YE chromosome and encodes for the streptogramin A acetyltransferase that confers resistance to streptogramins (50). In the current study, despite the presence or absence of the *vatF* gene, all *Yersinia* spp. isolates demonstrated similar susceptibility to quinupristin-dalfopristin (a streptogramin combination that had no interpretative AST criteria available). It has been previously reported that gram-negative organisms have a high level of intrinsic resistance to streptogramins, but elevated resistance may be due to simultaneous actions of several unknown mechanisms (51). The clinical relevance of streptogramin resistance in *Yersinia* is currently undetermined, but quinupristin-dalfopristin is not available for use in NZ.

One of the YE BT 2/3, ST12 genomes possessed a gene associated with tetracycline resistance (*tetA*) which correlated with phenotypic resistance to this antimicrobial. The study of Karlsson et al. (43) reported YE outbreak strains (BT 4) to possess a 5.7 kb plasmid that included the *tetB* (encoding an ABC transporter), as well as other resistance genes associated with *Pasteurellaceae*. The results from that study suggested that YE were able to acquire multi-antimicrobial and metal resistance genes through horizontal gene transfer (43). Intermediate susceptibility or resistance to tetracycline has also been reported for YE in other international studies (44, 52–54). As doxycycline is the first-choice antimicrobial for yersiniosis, increases in the prevalence of this gene in clinical *Yersinia* spp. would be of concern. This can be routinely monitored via the evaluation of genome-sequenced isolates.

The current study found that YP were susceptible to all NZ clinically relevant antimicrobials tested, which correlated with the observation of no known AMR genes within any of the YP genomes. This result is consistent with other international studies, but multi-drug resistant YP strains have been previously reported overseas (44, 55–57). Even though the frequency of multi-drug resistant YP is considered in the literature as low, the capacity of this species to acquire exogenous plasmids indicates a risk of the emergence of new multi-drug resistant YP strains (58).

This study did present some limitations such as the small number of participants and the reliance on information from patients directly and their memory recall. However, the study found that almost half of yersiniosis cases interviewed received antimicrobial treatment, with most prescriptions being directed therapy after a diagnosis was made. This study also confirms the appropriateness of current treatment guidelines for the management of yersiniosis in NZ, and that most primary care practitioners in the study areas follow these recommendations. Findings from phenotypic testing and genomic AMR assessment indicate that a low level of AMR was observed for *Yersinia* spp. in NZ compared to data from overseas studies. Internationally, multi-resistant *Yersinia* is reportedly rare among clinical isolates, but some studies have reported trends of increasing AMR to clinically relevant antimicrobials including trimethoprim-sulfamethoxazole and tetracycline (43, 46, 51, 59–63). Ongoing surveillance given the burden of yersiniosis and high rates of treatment in NZ is paramount to ensuring timely detection of emerging multi-drug resistance and to help devise evidence-informed interventions.

## ACKNOWLEDGMENTS

The authors acknowledge all the individuals who consented to participate in the case-control study and to Fiona for all her efforts with interviewing and Wendy Dallas-Katoa and Shevaun Paine for assistance with the questionnaire. The authors are also thankful for the NZ Ministry of Health for the *Yersinia* surveillance and typing in NZ and for the use of data collected within that process. Thank you to Jing Wang for her assistance with uploading genomes to NCBI. A final thanks to all the multiple staff within the Public Health Services for their support for case assessment and follow-up, those within diagnostic laboratories and at ESR who conducted laboratory work (sampling, typing, DNA extractions, and sequencing) and support staff that helped during the study.

This work was supported by the New Zealand Health Research Council (contract 20/847, 2020–2023) and the ESR Science Strategic Investment Fund of the Ministry of Business, Innovation, and Employment.

## AUTHOR AFFILIATIONS

[1]Institute of Environmental Science and Research (ESR), Christchurch Science Centre, Christchurch, Canterbury, New Zealand

[2]Institute of Environmental Science and Research, Kenepuru Science Centre, Porirua, Wellington, New Zealand

[3]Awanui Labs, Dunedin Hospital, Dunedin, Otago, New Zealand

[4]Institute of Environmental Science and Research, Wallaceville Science Centre, Upper Hutt, Wellington, New Zealand

## AUTHOR ORCIDs

Lucia Rivas  http://orcid.org/0000-0001-9666-495X
Brent Gilpin  http://orcid.org/0000-0002-8109-2271

## FUNDING

| Funder | Grant(s) | Author(s) |
| --- | --- | --- |
| Health Research Council of New Zealand | 20/847 | Lucia Rivas |
| | | Kristin Dyet |
| | | Jackie Wright |
| | | Brent Gilpin |
| Ministry for Business Innovation and Employment | | Lucia Rivas |
| | | Jenny Szeto |
| | | Juliet Elvy |
| | | Kristin Dyet |
| | | Jackie Wright |
| | | Ernest Williams |
| | | Brent Gilpin |

## AUTHOR CONTRIBUTIONS

Lucia Rivas, Conceptualization, Data curation, Formal analysis, Funding acquisition, Investigation, Methodology, Project administration, Writing – original draft, Writing – review and editing | Jenny Szeto, Data curation, Investigation, Methodology, Writing – original draft, Writing – review and editing | Juliet Elvy, Formal analysis, Methodology, Writing – original draft, Writing – review and editing | Kristin Dyet, Funding acquisition, Investigation, Methodology, Resources, Supervision, Writing – original draft, Writing – review and editing | Jackie Wright, Formal analysis, Funding acquisition, Methodology, Supervision, Writing – original draft, Writing – review and editing | Ernest Williams, Data curation, Formal analysis, Investigation, Methodology, Project administration, Writing – original draft, Writing – review and editing | Brent Gilpin, Conceptualization, Formal analysis, Funding acquisition, Project administration, Supervision, Writing – original draft, Writing – review and editing

## DATA AVAILABILITY

Raw sequence reads for all isolates analyzed in this study are available on National Centre for Biotechnology Information (NCBI) short read archive (SRA) with BioProject number PRJNA 1142067.

## ETHICAL APPROVAL

This study was approved by the New Zealand Southern Health and Disability Ethics (reference 21/STH/84).

## ADDITIONAL FILES

The following material is available online.

### Supplemental Material

**Supplemental Tables (Spectrum02751-24-s0001.xlsx).** Tables S1 and S2.

### Open Peer Review

**PEER REVIEW HISTORY (review-history.pdf).** An accounting of the reviewer comments and feedback.

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
