## [Reviewer comments · Microbiology Spectrum]

Microbiology Spectrum

Antimicrobial susceptibility and treatment of clinical cases of yersiniosis in Aotearoa New Zealand.

Lucia Rivas, Jenny Szeto, Juliet Elvy, Kristin Dyet, Jackie Wright, Ernest Williams, and Brent Gilpin

Corresponding Author(s): Brent Gilpin, Institute of Environmental Science and Research Ltd

Review Timeline:

Submission Date:	November 5, 2024
Editorial Decision:	January 22, 2025
Revision Received:	March 5, 2025
Accepted:	March 6, 2025

Editor: Mark Pandori

Reviewer(s): Disclosure of reviewer identity is with reference to reviewer comments included in decision letter(s). The following individuals involved in review of your submission have agreed to reveal their identity: Luis Augusto Nero (Reviewer #1)

Transaction Report:

DOI: <https://doi.org/10.1128/spectrum.02751-24>

Re: Spectrum02751-24 (Antimicrobial susceptibility and treatment of clinical cases of yersiniosis in Aotearoa New Zealand.)

Dear Dr. Brent J Gilpin:

Thank you for the privilege of reviewing your work. Below you will find my comments, instructions from the Spectrum editorial office, and the reviewer comments.

Revision Guidelines

Sincerely,
Mark Pandori
Editor
Microbiology Spectrum

Reviewer #1 (Comments for the Author):

In this study, Rivas et al investigated the characteristics of yersiniosis cases in New Zealand, followed by a laboratory study with the isolates to characterize their genomic and resistance profiles. The manuscript is well written and prepared, and considering the topic (yersiniosis, deep characterization of clinical cases and isolates - so difficult to do in some countries!). I just have some minor general comments and suggestions, to support a review and a novel analysis: 1) add further information about WGS

sequencing and bioinformatics procedures in methods; 2) please, provide some information about the reliability of Sensititre for checking the antimicrobial resistance profiles; 3) please, add some further bioinformatics analysis to demonstrate how the Yersinia isolates are similar/different from isolates from other countries... this is important to demonstrate how these isolates are related to other countries (and can be good hints of origin and sources). Congrats for the beautiful work!

Reviewer #2 (Comments for the Author):

I have read and reviewed the manuscript, "Antimicrobial susceptibility and treatment of clinical cases of yersiniosis in Aotearoa New Zealand" provided by Rivas et al. The manuscript describes an assessment of 148 cases of confirmed Yersinia spp. Infection. The assessment included bio typing, microbial use, resistance and symptomology. The manuscript is well-written and professionally presented. The use of terminology and statistics appears to be proper. The data that is presented has the potential to be useful to public health from a strategic perspective and to medical doctors, in a tactical mode. The manuscript, overall, appears to be a valid contribution to the knowledge base, within a chronological time frame. Critique of the manuscript follows:

1. It is understood that this is an assessment of infections in a local context. But given that we live in such a connected world in many respects, it begs the question of how the genomics and the resistances compare to the same organism in other parts of the world. Is the experience in this study comparable to other locations? The Discussion may be a good place to bring this up. Something along the lines of, "it seems fine here, and now, but ... is the potential for a change on the horizon?"
2. The Abstract on line 45 says, "...which was confirmed by whole genome...". I'm concerned with the wording. Whole genome may show data that has mutations or genes that correspond or are associated with phenotypes; but it can't really "confirm" them. For example, even in your study here, you saw TEM-1 but were noting some discordance in phenotype. Later in the manuscript, there is reference to "...correlates with WGS". I'm pretty sure I know what you mean, but, there must be some formal detail on how well these relevant genes have been shown to be associated with such phenotypes. References for those associations would formalize the use of such assertions.
3. Materials and Methods is very cursory with regard to your WGS work. Were all isolates sequenced? How many? What methods (lab bench)? Are the genomes available? Were they compared to other genomes worldwide or previously seen locally?
4. The manuscript describes a study that did not include medical records review, if I am reading this correctly. As such, patient memory was entirely relied upon. It really does seem like a LOT of detail was requested from memory. Was there any confirmation, or method to assess reliability of memory? This leads to the next point:
5. The Discussion should review weaknesses of the study: study size, the reliance on patient memory, etc.
6. There was an observation that YE BT 2/3 represented 20 of 25 hospitalized cases. This sounds interesting! Is this previously observed? A reference to any correlations between biotype and severity would be in order, in this manuscript.

Spectrum02751-24 - Antimicrobial susceptibility and treatment of clinical cases of yersiniosis in Aotearoa New Zealand.

4th March 2025

Dear Dr Pandori,

Thank you for the opportunity to revise the manuscript entitled “Antimicrobial susceptibility and treatment of clinical cases of yersiniosis in Aotearoa New Zealand (Spectrum02751-24)” for consideration for publication in the Spectrum Microbiology Journal.

We appreciated the comments received from both reviewers and have revised the manuscript accordingly. Please find below the changes made in response to each comment from the reviewers.

We trust that these changes are suitable for the reviewers and yourself and look forward to further communications from you.

Kind Regards,

Brent Gilpin PhD

Reviewer #1 comment	Author response and actions
In this study, Rivas et al investigated the characteristics of yersiniosis cases in New Zealand, followed by a laboratory study with the isolates to characterize their genomic and resistance profiles. The manuscript is well written and prepared, and considering the topic (yersiniosis, deep characterization of clinical cases and isolates - so difficult to do in some countries!). I just have some minor general comments and suggestions, to support a review and a novel analysis: 1) add further information about WGS sequencing and bioinformatics procedures in methods.	The work described in the manuscript was performed in line with a published case-control study (Rivas et al. 2024). The original manuscript referred to the case-control study for the methodologies for the whole genome sequencing and bioinformatic protocols. However, for clarity, these methods have now been included in the revised manuscript, specifically in the Method section: Bacterial isolates and genomic assessment (from Line 158)
2) please, provide some information about the reliability of Sensititre for checking the antimicrobial resistance profiles;	The following information and references have been added to the manuscript to address the reviewer’s comment: Line 187: This method provides a gold level standard for phenotypic susceptibility testing of bacterial isolates (Hughes et al. 2018; Li et al. 2020; Bhatnagar et al. 2023). Line 198: Results were only reported if the control strains were within range, as specified by EUCAST or CLSI.

3) please, add some further bioinformatics analysis to demonstrate how the Yersinia isolates are similar/different from isolates from other countries... this is important to demonstrate how these isolates are related to other countries (and can be good hints of origin and sources).
Congrats for the beautiful work!

We appreciate the reviewer's comments with respect to bioinformatic comparisons with isolates from other countries and agree that this exercise is worthwhile. It is an ongoing task for the authors, and we are currently working towards a separate manuscript detailing that work.

New Zealand has a unique yersiniosis epidemiology compared with other countries where yersiniosis is notifiable (Rivas et al. 2022). To address the reviewer's comment, it is apparent that providing more information on the yersiniosis epidemiology for New Zealand may be helpful in the current manuscript to provide context to the research. The current study (including isolates and interview responses) was aligned with a case-control study that focused on elucidating the sources of yersiniosis in New Zealand and included genomic comparisons with source isolates (Rivas et al. 2024). Although the genomic comparisons within the case-control study did not include international isolates, the findings show strong evidence that raw pork is a significant risk factor for the most predominate biotype (BT) of *Y. enterocolitica* causing illness, which is not as common in other countries that undertake surveillance. There are also differences in risk factors for other YE biotypes such as BT 1A which is not notifiable in many other countries. This BT is also very genomically heterogeneous as previously described.

The following information has been included in the introduction to provide context to yersiniosis in New Zealand:

Line 73: The predominance of YE biotype (BT) 4 is reported in other countries where surveillance and typing are available (Le Guern et al. 2016; ECDC, 2022). In NZ, a shift from a predominance from YE BT 4 to YE BT 2/3 amongst clinical cases occurred from 2012, and the reasons why this changed occurred remains unclear (Rivas et al. 2024). Also, YE BT 1A, a BT that is considered non-pathogenic due to the lack of classical YE virulence determinants is not notifiable in many countries yet is observed as the second most common YE BT in notified cases in NZ (Sihvonon et al. 2012;

	Campioni et al. 2014). A NZ case-control study on yersiniosis reported raw pork as a significant risk factor for exposure and infection to YE BT 2/3, but not YE BT 1A, indicating the etiology of different YE biotypes in NZ is complex (Rivas et al. 2024).
--	--

Reviewer #2 comments	Author response and actions
I have read and reviewed the manuscript, "Antimicrobial susceptibility and treatment of clinical cases of yersiniosis in Aotearoa New Zealand" provided by Rivas et al. The manuscript describes an assessment of 148 cases of confirmed Yersinia spp. Infection. The assessment included bio typing, microbial use, resistance and symptomology. The manuscript is well-written and professionally presented. The use of terminology and statistics appears to be proper. The data that is presented has the potential to be useful to public health from a strategic perspective and to medical doctors, in a tactical mode. The manuscript, overall, appears to be a valid contribution to the knowledge base, within a chronological time frame. Critique of the manuscript follows: 1. It is understood that this is an assessment of infections in a local context. But given that we live in such a connected world in many respects, it begs the question of how the genomics and the resistances compare to the same organism in other parts of the world. Is the experience in this study comparable to other locations? The Discussion may be a good place to bring this up. Something along the lines of, "it seems fine here, and now, but ... is the potential for a change on the horizon?"	New Zealand has a unique yersiniosis epidemiology compared with other countries where yersiniosis is notifiable (Rivas et al. 2021). The reviewer's comment is similar to one that was provided by reviewer 1 and it is apparent that providing some more information on the yersiniosis epidemiology for New Zealand may be helpful in the current manuscript to provide context to the research. The current study (including isolates and interview responses) was aligned with a case-control study that focused on elucidating the sources of yersiniosis in New Zealand and included genomic comparisons with source isolates (Rivas et al. 2024). Although the genomic comparisons within the case-control study did not include international isolates, the findings show strong evidence that raw pork is a significant risk factor for the most predominate biotype (BT) of Y. enterocolitica causing illness, which is not as common in other countries that undertake surveillance. There are also difference in risk factors for other YE biotypes such as BT 1A which is not notifiable in many other countries. This BT is also very genomically heterogeneous as previously described. The authors are undertaking further research that will be published separately. The following information has been included in the introduction to provide context to yersiniosis in New Zealand: Line 73: The predominance of YE biotype (BT) 4 is reported in other countries where surveillance and typing are available (Le Guern et al. 2016; ECDC, 2022). In NZ, a shift from a predominance from YE BT 4 to YE BT 2/3 amongst clinical cases occurred

	from 2012, and the reasons why this changed occurred remains unclear (Rivas et al. 2024). Also, YE BT 1A, a BT that is considered non-pathogenic due to the lack of classical YE virulence determinants is not notifiable in many countries, yet is observed as the second most common YE BT in notified cases in NZ (Sihvonen et al. 2012; Campioni et al. 2014). A NZ case-control study on yersiniosis reported raw pork as a significant risk factor for exposure and infection to YE BT 2/3, but not YE BT 1A, indicating the etiology of different YE biotypes in NZ is complex (Rivas et al. 2024). In addition, the following underlined text has also been included in the discussion: Line 405: Findings from phenotypic testing and genomic AMR assessment indicate there is a low level of AMR observed for Yersinia spp. in NZ compared with data from overseas studies. Internationally, multi-resistant Yersinia is reportedly rare amongst clinical isolates, but some studies have reported trends of increasing antimicrobial resistance to clinically relevant antimicrobials including trimethoprim-sulfamethoxazole and tetracycline (Capilla et al. 2003; Sihvonen et al. 2011; Karlsson et al. 2021; Marimon et al. 2017; Koskinen et al. 2022; Martins et al. 2022; Ye et al. 2023; Stevens et al. 2024). Ongoing surveillance given the burden of yersiniosis and high rates of treatment in NZ is paramount to ensuring timely detection of emerging multi-drug resistance and to help devise evidence-informed interventions.
2. The Abstract on line 45 says, "...which was confirmed by whole genome...". I'm concerned with the wording. Whole genome may show data that has mutations or genes that correspond or are associated with phenotypes; but it can't really "confirm" them. For example, even in your study here, you saw TEM-1 but were noting some discordance in phenotype. Later in the manuscript, there is reference to "...correlates with WGS". I'm pretty sure I know what you mean, but, there must be some formal detail on how well these	The authors agree with the reviewer on this comment. The following revisions were undertaken to address the reviewer's comment: Within the following sections (line numbers), the statements have been changed to read (as underlined): Abstract (Line 45): Whole genome sequencing (WGS) analysis showed very few (1 – 3) antimicrobial resistance (AMR) genes within the Yersinia genomes. The

relevant genes have been shown to be associated with such phenotypes. References for those associations would formalize the use of such assertions.	results support the current antimicrobial prescribing recommendation for the treatment of yersiniosis in NZ, and the utility of WGS to assess for AMR profiles in Yersinia spp. Results, Line 277: Whole genome sequence analysis resulted in very few (1 – 3) AMR genes detected amongst the Yersinia genomes (Table 4). Discussion, Line 405: Findings from phenotypic testing and genomic AMR assessment indicate the there is a low level of AMR observed for Yersinia spp. in NZ compared with data from overseas studies,... Statements within the discussion that referring to associations of phenotypic and antimicrobial resistance genes were reviewed but no further changes were made.
3. Materials and Methods is very cursory with regard to your WGS work. Were all isolates sequenced? How many? What methods (lab bench)? Are the genomes available? Were they compared to other genomes worldwide or previously seen locally?	As outlined for Reviewer 1, the methodology for the WGS component of the study has been added to the revised manuscript. This provides clarity as to the methods used including the number of isolates and genomes which are available in BioProject number where the WGS data can be accessed. As outlined in comment #1, no genomic comparisons were performed with international isolates due to the complex etiology of yersiniosis in NZ. Additional information was added to the introduction to provide greater context regarding yersiniosis in NZ (see response to comment #1).
4. The manuscript describes a study that did not included medical records review, if I am reading this correctly. As such, patient memory was entirely relied upon. It really does seem like a LOT of detail was requested from memory. Was there any confirmation, or method to assess reliability of memory? This leads to the next point:	This is a very good point by the reviewer and the authors did include a statement on the information relying on patient interviews only. To add to this as a study limitation, the following was also included in the results and discussion: Results, Line 205: The medium number of days between case notification and follow-up interviews with the questions specific for the current study was seven days. Discussion, Line 321: This collection of

	information relied on the case's memory recall and accuracy which can be deemed as a limitation to the study, especially with the time delay between case notification and interviewing (medium seven days).
5. The Discussion should review weaknesses of the study: study size, the reliance on patient memory, etc. In addition to the previous comment response that focus on reliance on patient memory, the following information has been added to the manuscript to address the study size:	As outlined in the comment prior, more information has been included in the manuscript on the reliance of patient memory. With respect to study size. The following information was included: Discussion, Line 315: The smaller number of case participants in the current study (n = 148) and the Swiss study (n = 66; Burnens et al. 1996) may explain why a statistical significance between case symptoms and YE BT was not observed in contrast to other studies (Finnish study; n = 295; Huovinen et al. (2010) and German study; n = 571; Rosner et al. (2013)) Line 399: This study did present some limitations such as the small number of participants and the reliance of information from patients directly and their memory recall. However, the study found that almost half of yersiniosis cases interviewed receive antimicrobial treatment, with most prescriptions as directed therapy after a diagnosis has been made.
6. There was an observation that YE BT 2/3 represented 20 of 25 hospitalized cases. This sounds interesting! Is this previously observed? A reference to any correlations between biotype and severity would be in order, in this manuscript.	To address this comment the following additional information has been added to the revised manuscript: Discussion, Line 304: However, it is of interest that the 80% of hospitalized cases in the current study were reported as YE BT 2/3, with severe abdominal pain commonly noted amongst the self-reporting comments. This type of abdominal pain (particularly of the right lower quadrant of the abdomen) is similar to the pain experienced with appendicitis but is instead associated with mesenteric adenitis which can be caused by YE (Helbling et al. 2017). A German study reported that diarrhea, abdominal pain and fever were more frequent in yersiniosis patients positive for serotype O:3 (which correlates with BT 4) compared with other non-O:3 (including

	O:9, O:5, 27 [which both correlate to BT 2/3], or non-specified serotypes). The high rate of hospitalization (27%; 153/571) was also significant associated with pain in the lower right abdomen (Rosner et al. 2013). In contrast, a Swiss study reported no significant difference in symptoms or hospitalization between patients with YE BT 1A (n = 23) or pathogenic YE (Biotype 2, 3 and 4; n = 43) (Burnens et al. 1996). The smaller number of case participants in the current study (n = 148) and the Swiss study (n = 66; Burnens et al. 1996) may explain why a statistical significance between case symptoms and YE BT was not observed in contrast to other studies (Finnish study; n = 295; Huovinen et al. (2010) and German study; n = 571; Rosner et al. (2013)).
--	---

References cited in the above response

- Bhatnagar AS, Machado MJ, Patterson L, Anderson K, Abelman RL, Bateman A, Biggs A, Bumpus-White P, Craft B, Howard M, LaVoie SP, Lonsway D, Sabour S, Schneider A, Snippes-Vagnone P, Tran M, Torpey D, Valley A, Elkins CA, Karlsson M, Brown AC. 2023. Antimicrobial Resistance Laboratory Network's multisite evaluation of the ThermoFisher Sensititre GN7F broth microdilution panel for antimicrobial susceptibility testing. *J Clin Microbiol* 61:e0079923.
- Burnens AP, Frey A, Nicolet J. 1996. Association between clinical presentation, biogroups and virulence attributes of *Yersinia enterocolitica* strains in human diarrhoeal disease. *Epidemiol Infect* 116:27-34.
- Campioni F, Falcao JP. 2014. Genotyping of *Yersinia enterocolitica* biotype 1A strains from clinical and non-clinical origins by pulsed-field gel electrophoresis. *Can J Microbiol* 60:419-24.
- Capilla S, Goñi P, Rubio MC, Castillo J, Millán L, Cerdá P, Sahagún J, Pitart C, Beltrán A, Gómez-Lus R. 2003. Epidemiological study of resistance to nalidixic acid and other antibiotics in clinical *Yersinia enterocolitica* O:3 isolates. *J Clin Microbiol* 41:4876-8.
- European Centre for Disease Prevention and Control (ECDC). 2022. Yersiniosis - annual epidemiology report for 2021. In ECDC (ed), Annual epidemiological report for 2021. ECDC, Stockholm.
- Helbling R, Conficconi E, Wyttenbach M, Benetti C, Simonetti GD, Bianchetti MG, Hamitaga F, Lava SA, Fossali EF, Milani GP. 2017. Acute nonspecific mesenteric lymphadenitis: more than "no need for surgery". *Biomed Res Int* 2017:9784565
- Hughes C, Ashhurst-Smith C, Ferguson JK. 2018. Gram negative anaerobe susceptibility testing in clinical isolates using Sensititre and Etest methods. *Pathology* 50:437-441.
- Huovinen E, Sihvonen LM, Virtanen MJ, Haukka K, Siitonen A, Kuusi M. 2010. Symptoms and sources of *Yersinia enterocolitica*-infection: a case-control study. *BMC Infect Dis* 10:122.
- Karlsson PA, Tano E, Jernberg C, Hickman RA, Guy L, Järhult JD, Wang H. 2021. Molecular characterization of multidrug-resistant *Yersinia enterocolitica* from foodborne outbreaks in Sweden. *Front Microbiol* 12:664665.

- Koskinen J, Ortiz-Martínez P, Keto-Timonen R, Joutsen S, Fredriksson-Ahomaa M, Korkeala H. 2022. Prudent antimicrobial use is essential to prevent the emergence of antimicrobial resistance in *Yersinia enterocolitica* 4/O:3 strains in pigs. *Front Microbiol* 13: 841841
- Le Guern AS, Martin L, Savin C, Carniel E. 2016. Yersiniosis in France: overview and potential sources of infection. *Int J Infect Dis* 46:1-7.
- Li T, Castañeda CD, Arick MA, 2nd, Hsu CY, Kiess AS, Zhang L. 2020. Complete genome sequence of multidrug-resistant avian pathogenic *Escherichia coli* strain APEC-O2-MS1170. *J Glob Antimicrob Resist* 23:401-403.
- Marimon JM, Figueroa R, Idigoras P, Gomariz M, Alkorta M, Cilla G, Pérez-Trallero E. 2017. Thirty years of human infections caused by *Yersinia enterocolitica* in northern Spain: 1985-2014. *Epidemiol Infect* 145:2197-2203.
- Martins BTF, Meirelles JL, Omori WP, Oliveira RR, Yamatogi RS, Call DR, Nero LA. 2022. Comparative genomics and antibiotic resistance of *Yersinia enterocolitica* obtained from a pork production chain and human clinical cases in Brazil. *Food Res Int* 152:110917.
- Rivas L, Strydom H, Paine S, Wang J, Wright J. 2021. Yersiniosis in New Zealand. *Pathogens* 10:191.
- Rivas L, Horn B, Armstrong B, Wright J, Strydom H, Wang J, Paine S, Thom K, Orton A, Robson B, Lin S, Wong J, Brunton C, Smith D, Cooper J, Mangalasseril L, Thornley C, Gilpin B. 2024. A case-control study and molecular epidemiology of yersiniosis in Aotearoa New Zealand. *J Clin Microbiol* 62:e0075424.
- Rosner BM, Werber D, Höhle M, Stark K. 2013. Clinical aspects and self-reported symptoms of sequelae of *Yersinia enterocolitica* infections in a population-based study, Germany 2009-2010. *BMC infectious diseases* 13:236-236.
- Sihvonen LM, Toivonen S, Haukka K, Kuusi M, Skurnik M, Siitonen A. 2011. Multilocus variable-number tandem-repeat analysis, pulsed-field gel electrophoresis, and antimicrobial susceptibility patterns in discrimination of sporadic and outbreak-related strains of *Yersinia enterocolitica*. *BMC Microbiol* 11:42
- Sihvonen LM, Jalkanen K, Huovinen E, Toivonen S, Corander J, Kuusi M, Skurnik M, Siitonen A, Haukka K. 2012. Clinical isolates of *Yersinia enterocolitica* biotype 1A represent two phylogenetic lineages with differing pathogenicity-related properties. *BMC Microbiol* 12:208.
- Stevens MJA, Horlbog JA, Diethelm A, Stephan R, Nüesch-Inderbinnen M. 2024. Characteristics and comparative genome analysis of *Yersinia enterocolitica* and related species associated with human infections in Switzerland 2019-2023. *Infect Genet Evol.*2024.105652:105652.
- Yue Y, Zheng J, Sheng M, Liu X, Hao Q, Zhang S, Xu S, Liu Z, Hou X, Jing H, Liu Y, Zhou X, Li Z. 2023. Public health implications of *Yersinia enterocolitica* investigation: an ecological modeling and molecular epidemiology study. *Infect Dis Poverty* 12:41.

Re: Spectrum02751-24R1 (Antimicrobial susceptibility and treatment of clinical cases of yersiniosis in Aotearoa New Zealand.)

Dear Dr. Brent J Gilpin:

Your manuscript has been accepted, and I am forwarding it to the ASM production staff for publication. Your paper will first be checked to make sure all elements meet the technical requirements. ASM staff will contact you if anything needs to be revised before copyediting and production can begin. Otherwise, you will be notified when your proofs are ready to be viewed.

Sincerely,
Mark Pandori
Editor
Microbiology Spectrum